# Sodium Glucose Co-Transporter 2 Inhibitor Ameliorates Autophagic Flux Impairment on Renal Proximal Tubular Cells in Obesity Mice

**DOI:** 10.3390/ijms21114054

**Published:** 2020-06-05

**Authors:** Kazuhiko Fukushima, Shinji Kitamura, Kenji Tsuji, Yizhen Sang, Jun Wada

**Affiliations:** Department of Nephrology, Rheumatology, Endocrinology and Metabolism, Okayama University Graduate School of Medicine, Dentistry and Pharmaceutical Sciences 2-5-1 Shikata-cho, Okayama-shi, Okayama 700-8558, Japan; p7927m3x@s.okayama-u.ac.jp (K.F.); gmd422036@s.okayama-u.ac.jp (K.T.); yizhensang@outlook.com (Y.S.); junwada@okayama-u.ac.jp (J.W.)

**Keywords:** sodium glucose co-transporter 2 inhibitor, mammalian target of rapamycin (mTOR), autophagy, obesity, multi lamellar body

## Abstract

Obesity is supposed to cause renal injury via autophagy deficiency. Recently, sodium glucose co-transporter 2 inhibitors (SGLT2i) were reported to protect renal injury. However, the mechanisms of SGLT2i for renal protection are unclear. Here, we investigated the effect of SGLT2i for autophagy in renal proximal tubular cells (PTCs) on obesity mice. We fed C57BL/6J mice with a normal diet (ND) or high-fat and -sugar diet (HFSD) for nine weeks, then administered SGLT2i, empagliflozin, or control compound for one week. Each group contained *N* = 5. The urinary *N*-acetyl-beta-d-glucosaminidase level in the HFSD group significantly increased compared to ND group. The tubular damage was suppressed in the SGLT2i–HFSD group. In electron microscopic analysis, multi lamellar bodies that increased in autophagy deficiency were increased in PTCs in the HFSD group but significantly suppressed in the SGLT2i group. The autophagosomes of damaged mitochondria in PTCs in the HFSD group frequently appeared in the SGLT2i group. p62 accumulations in PTCs were significantly increased in HFSD group but significantly suppressed by SGLT2i. In addition, the mammalian target of rapamycin was activated in the HFSD group but significantly suppressed in SGLT2i group. These data suggest that SGLT2i has renal protective effects against obesity via improving autophagy flux impairment in PTCs on a HFSD.

## 1. Introduction

Much attention was focused on the associations between obesity/metabolic syndrome (MetS) and kidney injury recently. In large-cohort studies, high body mass index was shown to be a risk factor for end-stage renal diseases independently compared to hypertension or diabetes [1]. Many researches were conducted on the mechanisms of kidney injury by obesity/MetS. In addition, autophagy deficiency is reported to be one of the renal injury causes [2]. Autophagy contributes to cellular homeostasis by promoting degradation and recycling of intracellular defective macromolecules and organelles, while autophagy deficiency leads to cellular death. Autophagy deficiency is observed in renal proximal tubular cells (PTCs) of obesity patients [2]. In addition, autophagy flux impairment by lysosomal dysfunction, inflammasome activation, and macrophage invasion was observed in PTCs of obesity mice [3]. Thus, autophagy deficiency is believed to be involved in kidney injury by obesity/MetS.

Recently, sodium glucose co-transporter 2 inhibitor (SGLT2i), belonging to a class of medications used for the treatment of type II diabetes mellitus via inhibiting reabsorption of glucose in PTCs and, therefore, lowering blood sugar, was reported to have renal protective effects in chronic kidney disease patients [4]. SGLT2i was reported to ameliorate glomerular hyperfiltration by normalization of tubuloglomerular feedback [5], as well as to improve the mitochondrial abnormalities in PTCs and to reduce the oxidative stress [6,7]. Effects of SGLT2i with respect to autophagy in the kidney were also researched; however, the results are controversial [8,9]. In this study, we investigated the therapeutic effect of SGLT2i on autophagy deficiency in PTCs of obesity mice.

## 2. Results

### 2.1. Effects of SGLT2i on Clinical Parameters in Mice

We measured physiological and metabolic parameters of four groups of mice as described in Figure 1 to determine the effect of SGLT2i for obesity-related renal injury (Table 1). Mice fed a high-fat and -sugar diet (HFSD) showed a significant increase in body weight compared to mice fed a normal diet (ND). HFSD-fed mice also showed a significant increase in systolic blood pressure and serum total cholesterol compared to ND-fed mice. Serum creatinine of HFSD-fed mice was also a little higher than that of ND-fed mice; however, blood urea nitrogen of HFSD-fed mice showed no significant increase compared to that of ND-fed mice. These data suggest that there was no significant difference in renal function between these two groups. SGLT2i administration significantly increased the amount of urine glucose in both ND-fed and HFSD-fed mice. Among HFSD-fed mice, blood glucose and serum total cholesterol levels in the SGLT2i-treated group were significantly lower than those in HFSD group mice. These results are consistent with reports of clinical researches which examined the lowering effects of blood glucose and serum total cholesterol [10]. However, there was no significant difference in bodyweight between the SGLT2i-treated group and the control group of HFSD-fed mice.

### 2.2. SGLT2i Decreased Lipid Accumulation in PTCs of Obesity Mice

To evaluate the kidney injury, we performed periodic acid–Schiff staining on kidney sections. No glomerular injury lesion or expansion was observed in HFSD-fed mice kidney (Figure 2A(c)), and interstitial inflammation was not noticeable. Vacuoles appeared in the PTCs of HFSD-fed mice (Figure 2A(g)), and SGLT2i significantly decreased these vacuoles (Figure 2B). These vacuoles were positive with toluidine blue staining, which detects phospholipids (Figure 2A(k,l)). PTCs of HFSD-fed mice were also positive with oil red O staining, which detects neutral triglycerides and lipids (Figure 2A, o), although less clear than with toluidine blue staining. These results suggest that SGLT2i could decrease lipid accumulation in PTC, consistent with other reports which examined the effect of a four-week administration of SGLT2i on the lipid accumulation of PTCs [11].

### 2.3. SGLT2i Decreased p62 Accumulation on PTCs in Obesity Mice

To evaluate autophagy flux in PTCs, we performed immunostaining of p62 on kidney sections. p62 binds to cytoplasmic bodies which are degraded by autophagy—such as damaged mitochondria, ubiquitinated proteins, and so on—and p62 accumulation is reported to increase in overnutrition diseases such as MetS and diabetes mellitus because of autophagy flux impairment [2]. p62 accumulations in PTCs of HFSD-fed mice were significantly increased compared with those in PTCs of ND-fed mice (Figure 3A,B). Among HFSD-fed mice, p62 accumulation in the SGLT2i-treated group was significantly lower than that in the control group (Figure 3B).

We also performed p62 immunostaining on LLC-PK1 cells. p62 accumulation was significantly increased when exposed to high glucose (25 mM) or palmitic acid (250 μM), and SGLT2i significantly decreased this p62 accumulation in LLC-PK1 cells exposed to high glucose or high glucose plus palmitic acid, but not significantly in those exposed to palmitic acid only (Figure 4).

These results suggest that SGLT2i promoted the degradation of p62 accumulation mainly via protecting PTCs from high glucose exposure.

### 2.4. SGLT2i Decreased Autolysosomes and Increased Autophagosomes in PTCs of Obesity Mice

Next, we performed ultrastructural analysis in PTCs by transmission electron microscopy to morphologically evaluate autophagy flux. Large-size residual bodies—around 3 to 6 μm in diameter, almost all of which were multi lamellar bodies (MLBs)—appeared in PTCs of HFSD-fed mice (Figure 5A). MLBs are reported to appear in PTCs of obesity mice as a result of lysosomal dysfunction, filled with phospholipids, and they have a lysosomal/autophagic origin (autolysosomes) [3]. The phospholipids in MLBs are supposed to originate from degraded organelles such as damaged mitochondria [2]. Most MLBs in this study had morphological characteristics of the autolysosome, with segregated organelles losing their distinct shape and acid phosphatase activity (black deposits) indicating that fusion with lysosomes occurred [12,13]. In addition to MLBs, mitochondrial damage (swelling, inner membrane disarrangement) also appeared in PTCs of HFSD-fed mice (Figure 5C). Damaged mitochondria are reported to appear in PTCs on obesity mice followed by reactive oxygen species generation or inflammasome activation [2,7]. Interestingly, the number and size of MLBs in HFSD-fed mice of the SGLT2i-treated group were significantly lower than those of the control group (Figure 5B). These data suggest that SGLT2i improved the lysosomal function of HFSD-fed mice. Moreover, mitophagosomes—autophagosomes of damaged mitochondria, morphologically characterized by double-membrane vacuoles delimitating non-degradative mitochondria [12,13]—appeared in PTCs of HFSD-fed mice treated with SGLT2i (Figure 5D). These results suggest that SGLT2i restored autophagy flux [14].

We also performed ultrastructural analysis in LLC-PK1 cells. Mitochondria damage was reported to occur in PTCs exposed to high glucose or palmitic acid [2,15]. Consistent with these reports, damaged mitochondria appeared together with a lot of residual bodies—most of which were showing the morphological characteristics of MLBs and/or autolysosomes—in LLC-PK1 cells exposed to high glucose or palmitic acid (Figure 6s–x). Larger MLBs than in other groups—around 5 µm in diameter—appeared in LLC-PK1 cells exposed to both high glucose and palmitic acid (Figure 6g), indicating the synergistic effect of high glucose and palmitate acid. Lipid droplets appeared in LLC-PK1 cells exposed to palmitic acid (Figure 6u,v), but not apparently in those exposed to high glucose only. Moreover, mitophagosomes appeared in SGLT2i-administered LLC-PK1 cells exposed to high glucose or high glucose plus palmitic acid (Figure 6y,z).

Taken together, SGLT2i ameliorates lysosomal dysfunction of obesity mice by mainly protecting PTCs from high glucose exposure, resulting in autophagy flux improvement.

### 2.5. SGLT2i Suppressed Mammalian Target of Rapamycin (mTOR) Activation in PTCs of Obesity Mice

Next, we examined the change in mTOR activation by SGLT2i. In overnutrition, mTOR is activated by glucose, insulin signal, or amino acids, before forming mTOR signaling complex 1 (mTORC1), which inhibits the transcription of lysosomal enzymes [16] and inhibits autophagosome formation [17]. mTORC1 also increases lipid synthesis and deposition in lysosome, resulting in lysosomal dysfunction [3,18,19]. In this study, we measured the expression of mTOR phosphorylated on serine 2448 (mTOR (pS2448)), which is predominantly contained in mTORC1 [20] and one of the indexes of mTOR activation. The expression of mTOR (pS2448) in the kidney was significantly higher in HFSD-fed mice compared to that in ND-fed mice (Figure 7). Among HFSD-fed mice, the expression of mTOR (pS2448) in the SGLT2i-treated group was significantly lower than that in the control group (Figure 7). These data suggest that SGLT2i could improve autophagy flux impairment in PTCs on obesity mice via suppression of mTOR activation.

### 2.6. SGLT2i Decreased the Amount of Urinary NAG of Obesity Mice Independent of Urinary Proteins

We evaluated urinary *N*-acetyl-β-d-glucosaminidase (NAG) for tubulointerstitial damage in obesity. Urinary NAG—a lysosomal hydrolase primarily originating from PTCs—is a marker of proximal tubular injury [21,22]. The amount of urinary NAG in HFSD-fed mice was significantly higher than that in ND-fed mice (Figure 8). Among HFSD-fed mice, urinary NAG in the SGLT2i-treated group was significantly lower than that in the control group (Figure 8A). However, we could not observe significant differences in urinary protein (Figure 8B), and there was no glomerular expansion or lesion in HFSD-fed mice of our study, as noted above (Figure 2A). It is important to note that urinary protein could affect the amount of urinary NAG, as the absorption of urinary protein into PTCs could cause nuclear factor-kappa B activation, inflammation, and tubular injury [23]. These data suggest that SGLT2i could directly protect PTCs independent of the effects on tubulo-glomerular feedback.

## 3. Discussion

Several renal protective mechanisms of SGLT2i were reported; however, the effect of SGLT2i on autophagy was not elucidated. In this study, we found that SGLT2i has a therapeutic effect on autophagy flux impairment in PTCs on obesity mice. These findings might contribute to elucidating the renal protective mechanism of SGLT2i in obesity patients.

We focused on lysosomal functions as a target of SGLT2i for improving autophagy flux impairment in PTCs. Lysosomes fuse with autophagosomes to form autolysosomes, and they play an important role in degradation process of autophagy. In obesity, lysosomal hydrolase is inactivated by the pH increase caused by lipid accumulation in the lysosome, resulting in impairment of lysosomal capacity and autophagy flux [3]. Autolysosomes, the degradation capacity of which is impaired by lipid accumulation, are reported to form MLBs in obesity mice [3]. Our findings, i.e., the decrease of autolysosomes and the increase of mitophagosomes after SGLT2i administration, suggest the therapeutic effect of SGLT2i on lysosomal dysfunction and autophagy flux impairment in PTCs. We considered that these effects are due to the suppression of the activated mTOR as the result of the PTC’s decreased exposure of glucose. Activated mTOR forms mTORC1 and inhibits the transcription of lysosomal enzymes [16], as well as increases lipid deposition in lysosome, resulting in lysosomal dysfunction [3,18,19]. In this study, SGLT2i increased the amount of urinary glucose—thus decreasing the reabsorption of glucose into PTCs—and decreased the amount of serum glucose, suggesting the PTC’s decreased exposure of glucose, both in apical and basal sides. Moreover, we demonstrated that SGLT2i suppressed mTOR activation in obesity mice. Furthermore, our in vitro study showed that SGLT2i significantly improved autophagy flux of PTCs exposed to high glucose. These data suggest that the decrease of glucose influx into PTCs by SGLT2i could directly affect mTOR activation in PTCs.

Our study has at least two limitations. One is that we could not definitively distinguish SGLT2i’s effect for glucose and lipid metabolism in protecting PTCs; SGLT2i might have protected PTCs by also improving the lipid metabolism, resulting in amelioration of lipotoxicity in PTCs. In our in vivo study, SGLT2i administration at least showed no significant difference in bodyweight in obesity mice, indicating no significant difference in body fat. Furthermore, our in vitro study suggests that PTC’s protective effect of SGLT2i by inhibiting glucose absorption is independent of lipid exposure. The other limitation is that SGLT2i might have effects on autophagy of PTCs via other mechanisms. SGLT2i is reported to have effects on various factors related to autophagy, such as serum insulin, insulin-like growth factor-1, and sirtuin 1 [17,24,25]. Further studies are needed to elucidate more deeply how SGLT2i affects the autophagy pathway in PTCs.

It is important to note that SGLT2i showed protective effects of PTCs in obesity mice showing no significant increase in serum glucose compared to control mice in this study. Similarly, there is a report which showed SGLT2i’s protective effect of PTC’s mitochondria of obesity mice, which showed no significant elevation of serum glucose [7]. These findings indicate that over-intake of sugar might cause glucotoxicity of PTCs in obesity mice even if there is no increase of serum glucose, and urinary glucose might have an important role in PTC’s glucotoxicity. Unlike most types of cells which take in glucose only for their energy consumption, PTCs must reabsorb much urinary glucose for the whole body’s survival and, thus, might be under stronger exposure than other types of cells. In fact, reabsorption of urinary glucose in proximal tubules plays an important role in the progression of tubular injuries in diabetic nephropathy [26]. Further studies are needed to elucidate deeply whether urinary glucose reabsorption could worsen PTC’s injuries in obesity.

There is also a report which concluded that SGLT2i has no sufficient effect on autophagy in PTCs. Tanaka et al. reported that autophagy insufficiency in the tubular cells of diabetic mice was ameliorated by food-intake restriction, but not by SGLT2i [9]. They reported that the results were because the effect of SGLT2i on calorie balance is weaker than that of food-intake restriction; in their study, the amount of food intake and the bodyweight increase of the SGLT2i-treated group were greater than those in the control group. This might be because of an appetite increase via the glucose-lowering effect of SGLT2i. The change in the bodyweight by SGLT2i administration in their study was contrary to that in clinical trials, where SGLT2i significantly decreased the bodyweight in humans [10]. These reports suggest that free feeding of mice could mask the effect of SGLT2i on autophagy, and that the adjustment of food intake is needed to accurately evaluate the autophagy activity in in vivo experiments.

In conclusion, our study suggests that SGLT2i might ameliorate proximal tubular injuries in obesity via improving autophagy flux impairment.

## 4. Materials and Methods

### 4.1. Animal Models

This study was approved by the animal care and use committee of Okayama University (OKU-2018206). We used five-week-old male mice with a C57BL/6J genetic background (Charles River, Cologne, Germany). The mice were fed with normal diet with the composition of 13 kcal% fat (Cat#: MF, Oriental Yeast Co., ltd, Tokyo, Japan) or high-fat diet with the composition of 40 kcal% fat and 40 kcal% sucrose (Cat#: D12327, Research Diets INC, New Brunswick, USA). After nine weeks of feeding, we administrated 10.0 mg/kg SGLT2i (empagliflozin provided from Boehringer-Ingelheim, Ingelheim, Germany) dissolved with 0.5% hydroxypropyl methylcellulose (HPMC) or 0.5% HPMC (as control) orally for one week. After that, the mice were sacrificed, and the serum plasma, urine (collected for 24 h), and kidneys were obtained.

### 4.2. Cell Culture

LLC-PK1 cells (an epithelial cell line originated in proximal tubular cells of porcine kidney) were cultured in Medium 199 (M4530; Sigma-Aldrich, St. Louis, MO, USA) supplemented with 10% fetal bovine serum, 100 U/mL penicillin, and 100 mg/mL streptomycin at 37 ℃ in 5% CO_2_. When 80% confluent, the cells were serum-starved by culture in 0.1% fetal bovine serum for 24 h, then exposed to 5.5 mM glucose or 25 mM glucose in the presence or absence of 500 nM SGLT2i for up to 48 h before harvesting. Then, 24 h after culture initiation, the medium was changed and 250 μM palmitic acid (P0500; Sigma-Aldrich, St. Louis, MO, USA) dissolved in 99% chloroform (Wako Pure Chemical Industries, Ltd., Osaka, Japan) or 99% chloroform (as control) was added.

### 4.3. Histopathological Analysis

Sections were cut to 4 μm thickness from 10% formalin-fixed paraffin-embedded kidney samples. These sections were used for periodic acid–Schiff staining. Frozen sections that were cut to 8 μm thickness were used for oil red O staining of neutral lipid (cholesterol esters and triglycerides) deposits and immunostaining. For toluidine blue staining of phospholipid deposits, kidney specimens and LLC-PK1 cells were fixed with 2.5% glutaraldehyde.

For immunofluorescence microscopy, frozen sections (4 μm) and LLC-PK1 cells were stained using anti-p62 antibody (Cat #: PM066, Medical & Biological Laboratories Co., Ltd., Nagoya, Japan), anti-aquaporin 1 antibody (Cat #: AQP-001, Alomone labs, Jerusalem, Israel), anti-SGLT2 antibody (sc-47402; Santa Cruz Biotechnology, Dallas, TX, USA), and/or 4′,6-diamidino-2-phenylindole (DAPI). Sections were examined using an Olympus FSX100 biological microscope (Olympus Corporation, Tokyo, Japan). In a kidney histological quantitation, a minimum of six non-overlapping images of cortex per kidney were observed under 400× magnification.

For electron microscopy, kidney specimens and LLC-PK1 cells were fixed with 2.5% glutaraldehyde and observed using a transmission electron microscope (H- 7650, Hitachi, Tokyo, Japan).

Quantitative computer-assisted image analysis was performed using a digital image-analyzing software, ImageJ (available at http://rsbweb.nih.gov/ij/index.html; National Institutes of Health).

### 4.4. Western Blot Analysis

Proteins were extracted using a lysis buffer (Cat #: 89900, Thermo Fisher Scientific, Inc., Waltham, MA, USA) containing a protease inhibitor (Cat #: G652A, Promega Corporation, Fitchburg, WI, USA) and a phosphatase inhibitor (Cat #: ab201113, Abcam plc., Cambridge, UK), then quantified using a bicinchoninic acid (BCA) protein assay kit (Thermo Fisher Scientific Inc., Waltham, MA, USA). Equal amounts of protein were separated using 10% sodium dodecyl sulfate–polyacrylamide gel electrophoresis and electrophoretically transferred onto nitrocellulose membranes, which were blocked with 1% bovine serum albumin (Cat #: A7888-50G, Sigma-Aldrich Co. LLC, St. Louis, MO, USA) for 30 min. The membranes were incubated with the following primary antibodies overnight at 4 °C: anti-phospho mTOR (1:1,000, Cat #: 2971, Cell Signaling Technology, Danvers, MA, USA) and anti-glyceraldehyde 3-phosphate dehydrogenase (GAPDH; 1:10,000, Cat #: 2118, Cell Signaling Technology, Danvers, MA, USA). They were incubated with a horseradish peroxidase (HRP)-conjugated secondary antibody (1:2,000, Cat#: 170-6515, Bio-Rad Laboratories, Inc., Hercules, CA, USA) for 1 h. The membranes were extensively washed in phosphate-buffered saline with Tween-20, and antigen–antibody complexes were visualized by chemiluminescence using an enhanced chemiluminescence (ECL) kit (ECLTM Prime Western Blotting Reagents, GE Healthcare, Chicago, IL, USA).

### 4.5. Statistical Analysis

Results are presented as the mean ± standard deviation. Differences were evaluated by two-way ANOVA followed by Tukey–Kramer test (JMP^®^ 13.2, SAS Institute INC, Cary, USA). Statistical significance was accepted at the *p* < 0.05 level.

## Figures and Tables

**Figure 1 ijms-21-04054-f001:**
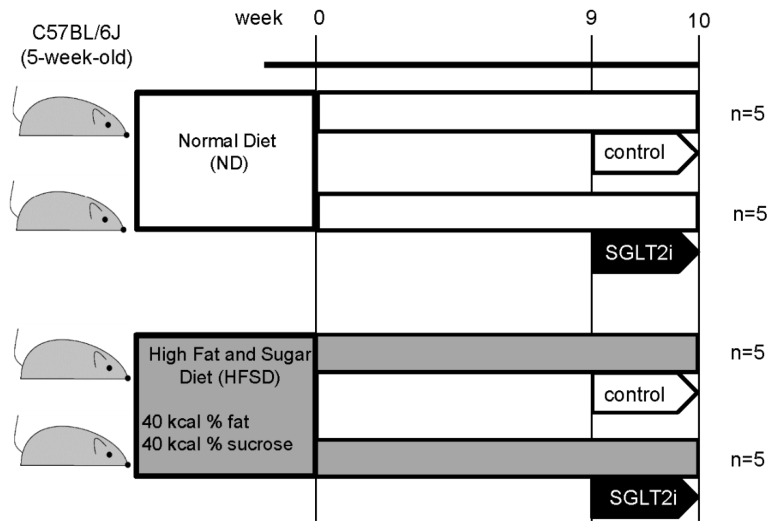
Physiological and metabolic parameters in normal diet (ND)-fed group and high-fat and -sugar diet (HFSD)-fed group. Results are presented as the mean ± standard deviation. Differences were evaluated by two-way ANOVA followed by Tukey–Kramer test; * *p* < 0.05 vs. control ND, # *p* < 0.05 vs. control HFSD. SGLT2i: empagliflozin.

**Figure 2 ijms-21-04054-f002:**
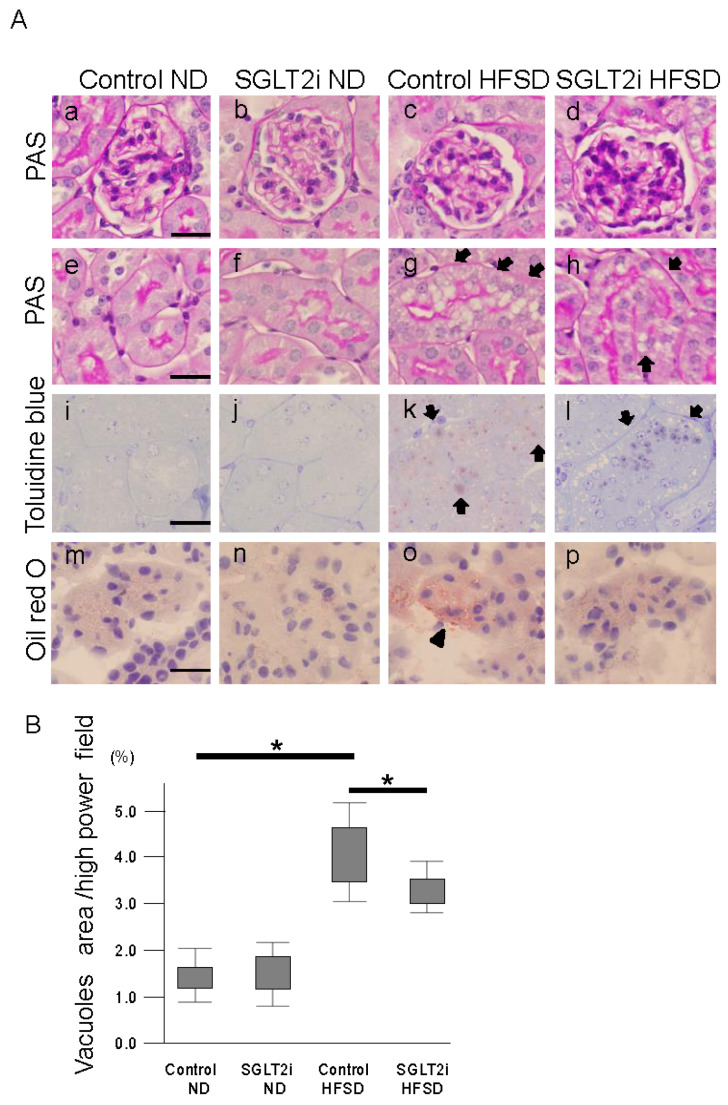
SGLT2 inhibitor (SGLT2i) decreased lipid accumulation in renal proximal tubular cells (PTCs). (**A**) Light microscope analysis of kidney sections. Periodic acid–Schiff (PAS) staining of (**a**–**d**) glomerular and (**e**–**h**) proximal tubules; (**i**–**l**) toluidine blue staining of kidney sections; (**m**–**p**) oil red O staining of kidney sections. (**B**) Quantitative analysis of vacuole area in PTCs; arrow: vacuoles in PTCs; arrowhead: oil red O-positive area. Results are presented as the mean ± standard deviation. Differences were evaluated by two-way ANOVA followed by Tukey–Kramer test (* *p* < 0.05); bars: 25 µm (**A**). SGLT2i: empagliflozin.

**Figure 3 ijms-21-04054-f003:**
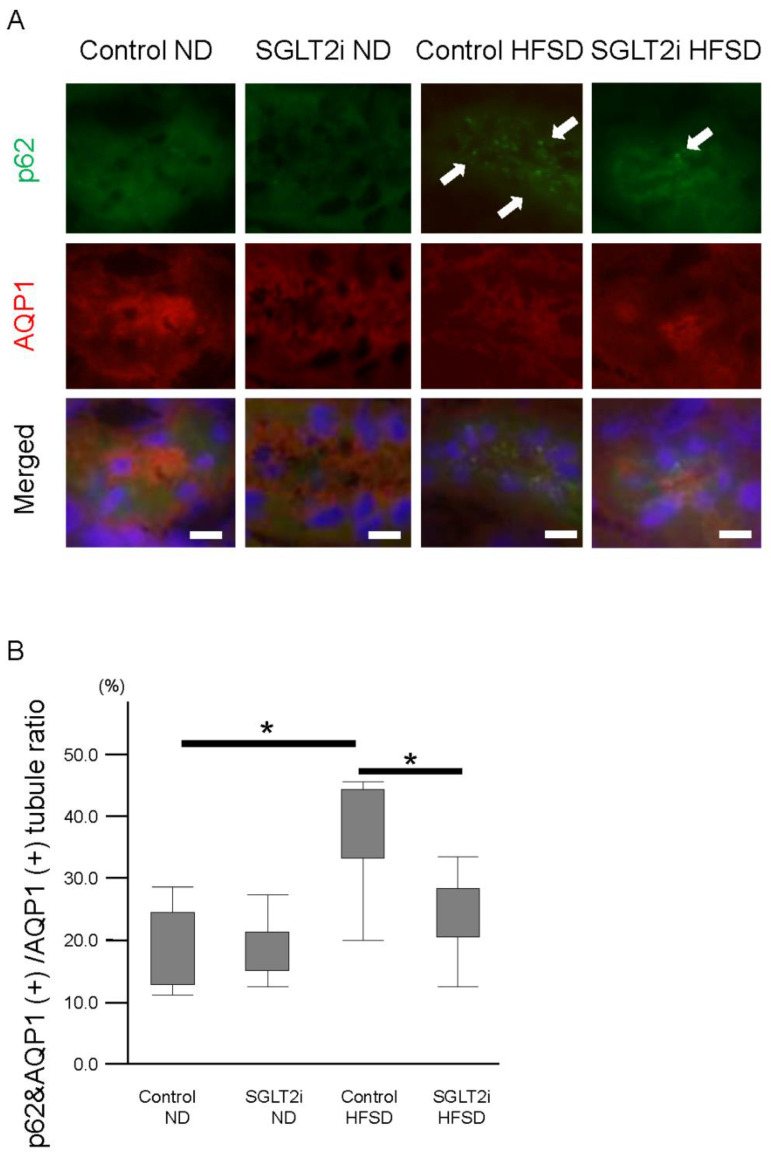
SGLT2 inhibitor (SGLT2i) decreased p62 accumulation on renal proximal tubular cells in obesity mice. (**A**) Immunofluorescence analysis for p62 accumulations; arrow: p62 accumulation. Proximal tubular cells were identified by aquaporin 1 (AQP1) staining. (**B**) Quantitative analysis of p62-positive proximal tubules. Results are presented as the mean ± standard deviation. Differences were evaluated by two-way ANOVA followed by Tukey–Kramer test (* *p* < 0.05); bars: 10 µm. SGLT2i: empagliflozin.

**Figure 4 ijms-21-04054-f004:**
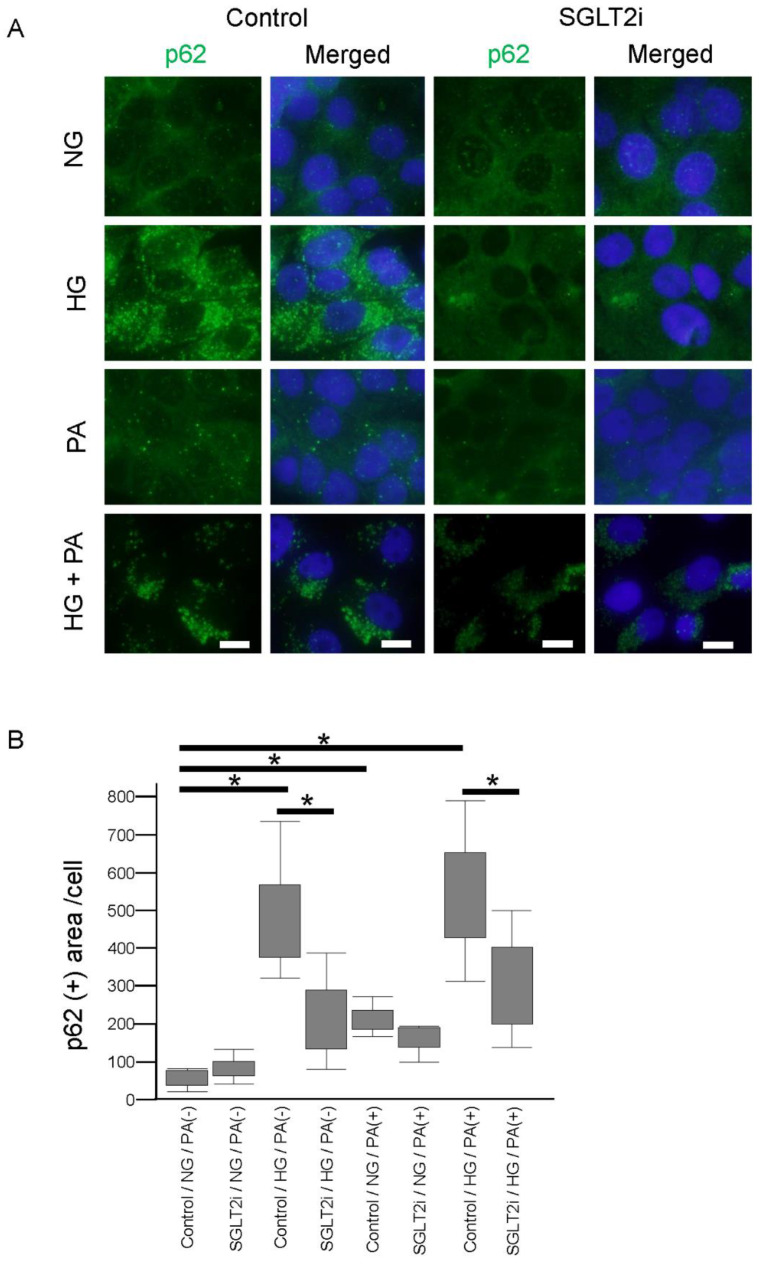
SGLT2 inhibitor (SGLT2i) decreased p62 accumulation on LLC-PK1 cells (**A**) Immunofluorescence analysis for p62 accumulation. (**B**) Quantitative analysis of p62-positive area per cell. Results are presented as the mean ± standard deviation. Differences were evaluated by two-way ANOVA followed by Tukey–Kramer test (* *p* < 0.05); bars: 10 µm (normal glucose: NG, high glucose: HG, palmitic acid: PA). SGLT2i: empagliflozin.

**Figure 5 ijms-21-04054-f005:**
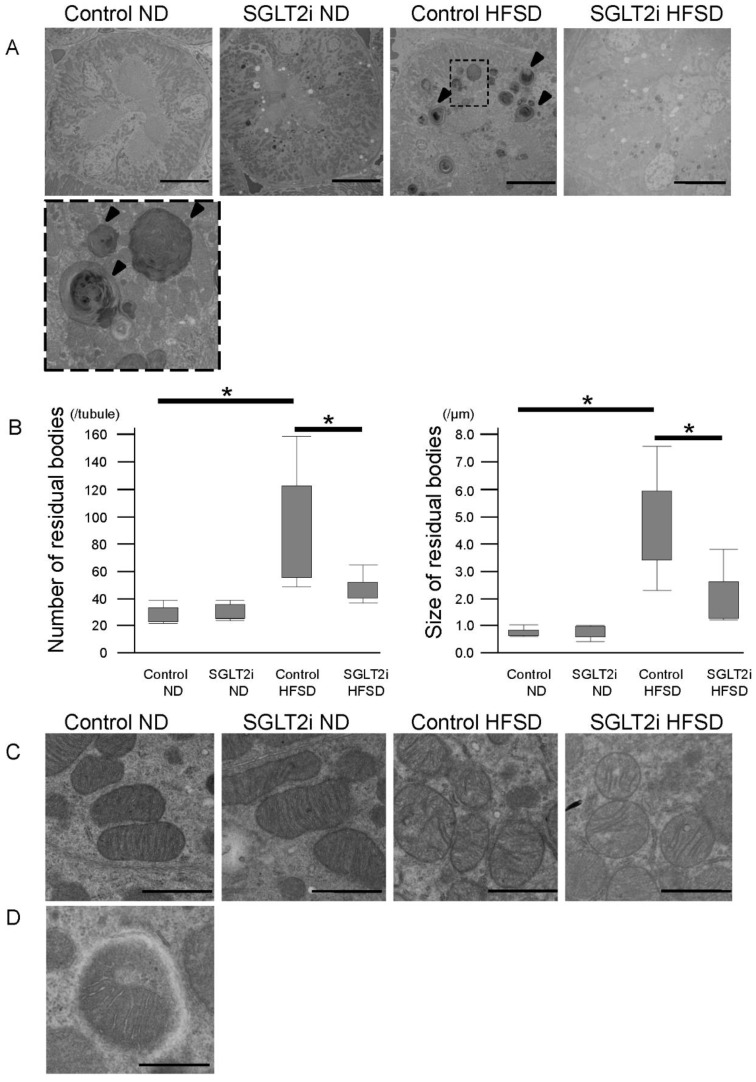
SGLT2 inhibitor (SGLT2i) decreased multi lamellar bodies (MLBs) and increased autophagosomes in renal proximal tubular cells (PTCs) of obesity mice. (**A**) PTCs in transmission electron microscopy images; arrowhead: MLBs; bars: 10 µm. (**B**) Quantitative analysis of residual bodies in PTCs. Results are presented as the mean ± standard deviation. Differences were evaluated by two-way ANOVA followed by Tukey–Kramer test (* *p* < 0.05). (**C**) Mitochondria in PTCs of transmission electron microscopy images; bars: 1.0 µm. (**D**) Mitophagosomes in PTCs of HFSD-fed mice treated with SGLT2i; bars: 500 nm. SGLT2i: empagliflozin.

**Figure 6 ijms-21-04054-f006:**
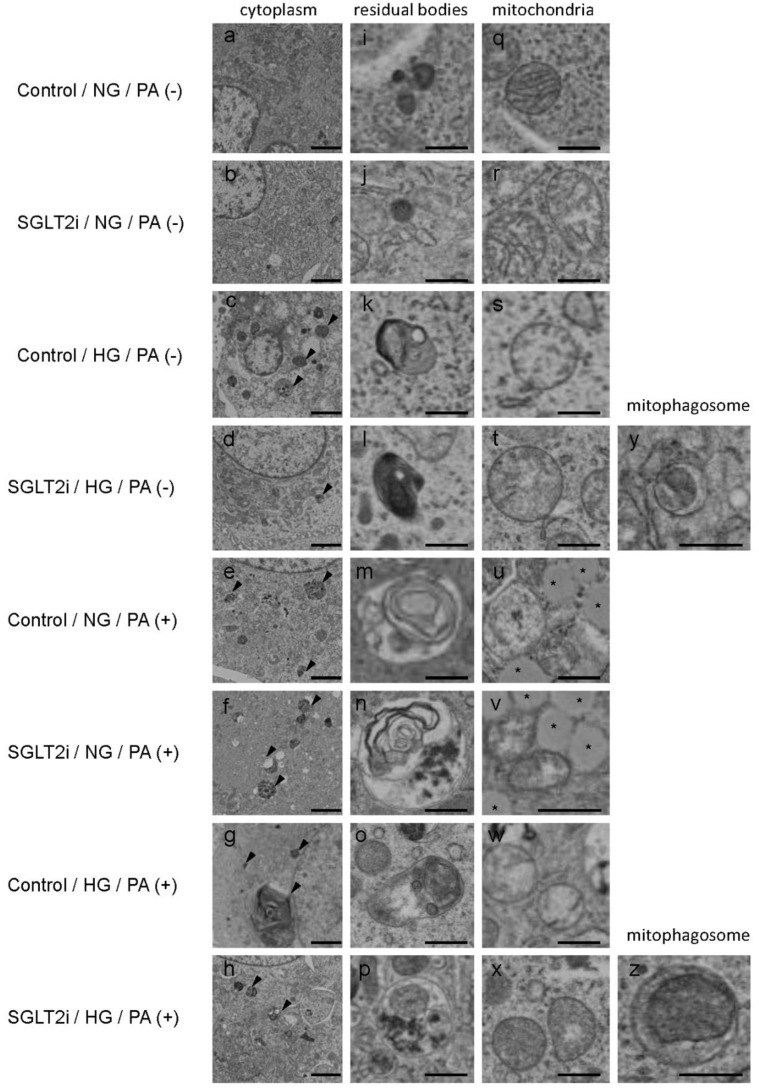
SGLT2 inhibitor (SGLT2i) increased autophagosomes of damaged mitochondria in LLC-PK1 cells. (**a**–**h**) Cytoplasm of LLC-PK1 cell; arrowhead: residual bodies; bars: 2.0 µm. (**i**–**p**) Representative images of residual bodies; bars: 500 nm. (**q**–**x**) Representative images of mitochondria; asterisk: lipid droplets; bars: 500 nm. (**y**,**z**) Representative images of mitophagosomes; bars: 500 nm. SGLT2i: empagliflozin.

**Figure 7 ijms-21-04054-f007:**
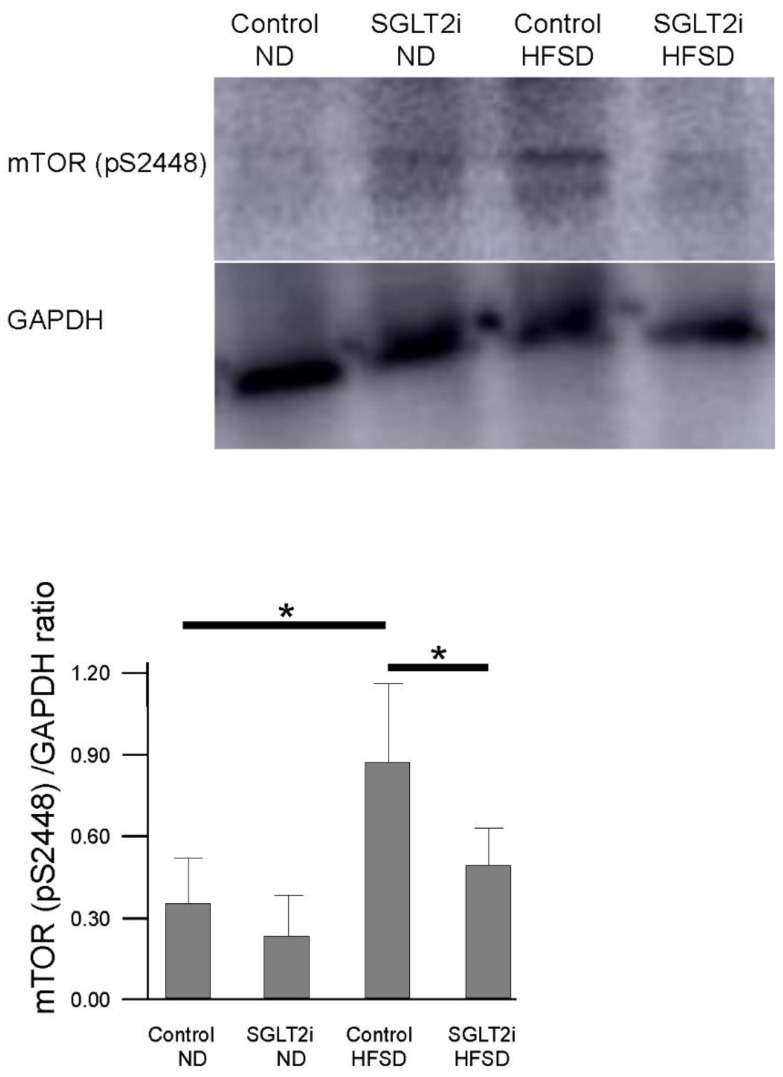
SGLT2 inhibitor (SGLT2i) suppressed mammalian target of rapamycin (mTOR) activation in renal proximal tubular cells of obesity mice. Western blotting analysis for mTOR phosphorylated on serine 2448 (mTOR (pS2448) of the kidney. Results are presented as the mean ± standard deviation. Differences were evaluated by two-way ANOVA followed by Tukey–Kramer test (* *p* < 0.05). SGLT2i: empagliflozin.

**Figure 8 ijms-21-04054-f008:**
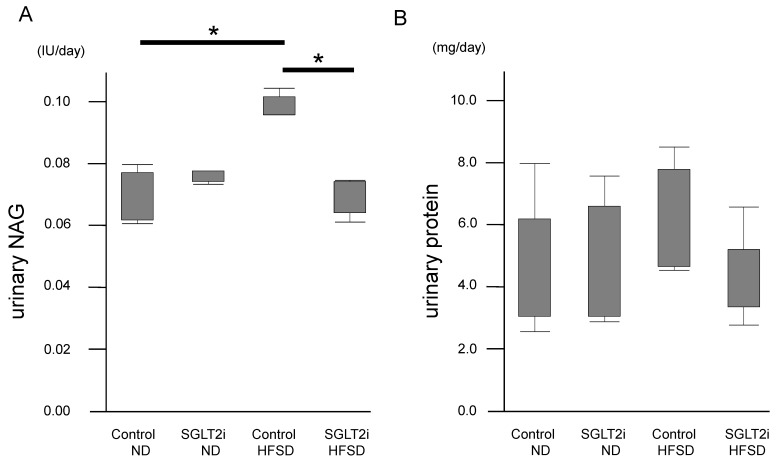
SGLT2 inhibitor (SGLT2i) decreased the amount of urinary *N*-acetyl-beta-d-glucosaminidase (NAG) independent of urinary proteins. The amount of (**A**) urinary NAG and (**B**) protein. The urine was collected for 24 h one day before sacrifice. Results are presented as the mean ± standard deviation. Differences were evaluated by two-way ANOVA followed by Tukey–Kramer test (* *p* < 0.05). SGLT2i: empagliflozin.

**Table 1 ijms-21-04054-t001:** Physiological and metabolic parameters in normal diet (ND)-fed group and high-fat and -sugar diet (HFSD)-fed group. Results are presented as the mean ± standard deviation. Differences were evaluated by two-way ANOVA followed by Tukey–Kramer test; * *p* < 0.05 vs. control ND, ^#^
*p* < 0.05 vs. control HFSD. SGLT2i—sodium glucose co-transporter 2 inhibitor: empagliflozin.

	Control ND	SGLT2i ND	Control HFSD	SGLT2i HFSD
Body Weight (g)	24.40 ± 1.37	26.44 ± 1.25	31.45 ± 1.56 *	29.90 ± 0.85
Systolic blood pressure (mmHg)	101.4 ± 9.0	101.0 ± 11.5	120.6 ± 7.7 *	109.2 ± 4.1
Diastolic blood pressure (mmHg)	76.0 ± 12.0	65.8 ± 12.8	73.0 ± 11.5	75.8 ± 5.3
Urine glucose (mg/day)	7.8 ± 8.8	294.2 ± 65.6 *	1.1 ± 1.6	269.1 ± 39.3 ^#^
Blood glucose (mg/dL)	256.6 ± 24.0	268.8 ± 37.1	308.8 ± 61.5	204.6 ± 27.8 ^#^
Total cholesterol (mg/dL)	79.8 ± 3.2	89.6 ± 8.3	138.2 ± 16.8 *	113.0 ± 9.3 ^#^
Triglycerides (mg/dL)	35.8 ± 8.8	31.4 ± 6.2	38.4 ± 24.1	25.0 ± 6.6
Glycoalbumin (%)	2.48 ± 0.69	2.74 ± 0.42	3.32 ± 0.42	2.80 ± 0.95
Serum creatinine (mg/dL)	0.088 ± 0.008	0.088 ± 0.013	0.120 ± 0.027 *	0.112 ± 0.013
Blood urea nitrogen (mg/dL)	23.54 ± 1.86	21.42 ± 1.85	17.18 ± 6.42	15.72 ± 3.33

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
