# Peer review of "Sodium Glucose Co-Transporter 2 Inhibitor Ameliorates Autophagic Flux Impairment on Renal Proximal Tubular Cells in Obesity Mice"

_ijms, 2020, doi:10.3390/ijms21114054_

Round 1
Reviewer 1 Report
This manuscript, original article-type, written by Dr. Kazuhiko Fukushima et al, and with the title of “Sodium glucose co-transporter 2 inhibitor ameliorates autophagic flux impairment on renal proximal tubular cells in obesity mice” focuses on the analysis of the therapeutic effect of empagliflozin (a SGLT2 inhibitor) in the autophagic deficiency in the kidneys of obesity mice.
Empagliflozin is a drug approved for the treatment of type 2 diabetes. It belongs to the gliflozin class (SGLT2 inhibitors). SGLT2 is a transporter found almost exclusively in the proximal tubules of the kidney and it is responsible for most of the glucose reabsorption from the glomerular filtrate into the blood. Autophagy is a catabolic process that results in the autophagosomic-lysosomal degradation of bulk cytoplasmic contents, abnormal protein aggregates, and excess or damaged organelles. Autophagy is a key regulatory pathway to preserve lipid homeostasis and insulin sensitivity under metabolic stress conditions. Dysfunction of autophagy is associated with obesity and type 2 diabetes.
The authors found that Empagliflozin had several effects: (1) Lowered the blood glucose and total serum cholesterol. (2) Decreased the lipid accumulation in the proximal tubules of the kidney. (3) Reduced the p62 accumulation in the proximal tubules. (4) Decreased the accumulation of multilamellar bodies in the proximal tubules (i.e. improved the lysosomal function). (5) Suppressed the mTOR activation. (6) Decreased the amount of urinary NAG (a marker of proximal tubular injury). The authors concluded that Empagliflozin has a renal protective effect against obesity by improving the autophagy impairment.
This manuscript is well written, it is easy to read, the results and their interpretation is correct, there are enough tables and figures and the references are correct.
Minor comments:
(1) Prior to publication in this Journal the authors may explain why they did not performed immunohistochemistry and opted for electron microscopy for the analysis of the impairment of autophagy.
(2) I think that they should write in the abstract the name of the SGLT2 inhibitor that they are testing and the number of mice that were used in the experiments.
(3) The authors may comment if they saw an increase of macrophages in the kidneys, next to the proximal tubules.
Author Response
Thank you for your kind review. We appreciate for your point out suggestion. Our manuscript would be more strength according to your suggestion.
We revised our manuscript according to review suggestion. The added sentences are underlined.
Thank you again for your kindness.
Reply for review:
Minor comments:
- Prior to publication in this Journal the authors may explain why they did not performed immunohistochemistry and opted for electron microscopy for the analysis of the impairment of autophagy.
Thank you for your suggestion. There are many studies on autophagy with the data of immunostaining. Regarding in this study, the electron microscopy analysis could show morphologically distinguish between autophagosome and autolysosome, and could demonstrate mitophagy targeting damaged mitochondria. Therefore, we showed electron microscopy data in this study. We added the explanation in the part of Results section in the manuscript. The added sentences are underlined.
- I think that they should write in the abstract the name of the SGLT2 inhibitor that they are testing and the number of mice that were used in the experiments.
According to your suggestion, we revised at the part of abstract section in the manuscript and Figure1. The revised parts are underlined.
(3) The authors may comment if they saw an increase of macrophages in the kidneys, next to the proximal tubules.
The interstitial inflammation is not so noticeable. We revised the description at the part of Results section in the manuscript.

Reviewer 2 Report
Dear Sirs
I enjoyed reading your manuscript. It has a scientifically merit and sounds good to me. Therefore , I recommended acceptance. Congratulation!
Author Response
Reply for reviewer 2:
I enjoyed reading your manuscript. It has a scientifically merit and sounds good to me. Therefore , I recommended acceptance. Congratulation!
Thank you for your evaluation and review.
